# *ACE2* and a Traditional Chinese Medicine Formula NRICM101 Could Alleviate the Inflammation and Pathogenic Process of Acute Lung Injury

**DOI:** 10.3390/medicina59091554

**Published:** 2023-08-26

**Authors:** Cheng-Han Lin, Yi-Ju Chen, Meng-Wei Lin, Ho-Ju Chang, Xin-Rui Yang, Chih-Sheng Lin

**Affiliations:** 1Department of Biological Science and Technology, National Yang Ming Chiao Tung University, Hsinchu 30068, Taiwan; a0975273923@gmail.com (C.-H.L.); tam13053@gmail.com (Y.-J.C.); imhrchang@gmail.com (H.-J.C.); yllas0315@gmail.com (X.-R.Y.); 2Center for Intelligent Drug Systems and Smart Bio-Devices (IDS2B), National Yang Ming Chiao Tung University, Hsinchu 30068, Taiwan; 3Department of Biological Science and Technology, National Chiao Tung University, Hsinchu 30068, Taiwan

**Keywords:** angiotensin-converting enzyme type II (ACE2), acute lung injury (ALI), animal model, traditional Chinese medicine

## Abstract

COVID-19 is a highly transmittable respiratory illness caused by SARS-CoV-2, and acute lung injury (ALI) is the major complication of COVID-19. The challenge in studying SARS-CoV-2 pathogenicity is the limited availability of animal models. Therefore, it is necessary to establish animal models that can reproduce multiple characteristics of ALI to study therapeutic applications. The present study established a mouse model that has features of ALI that are similar to COVID-19 syndrome to investigate the role of ACE2 and the administration of the Chinese herbal prescription NRICM101 in ALI. Mice with genetic modifications, including overexpression of human ACE2 (K18-hACE2 TG) and absence of ACE2 (mACE2 KO), were intratracheally instillated with hydrochloric acid. The acid intratracheal instillation induced severe immune cell infiltration, cytokine storms, and pulmonary disease in mice. Compared with K18-hACE2 TG mice, mACE2 KO mice exhibited dramatically increased levels of multiple inflammatory cytokines (IL-6 and TNF-α) in bronchoalveolar lavage fluid, histological evidence of lung injury, and dysregulation of MAPK and MMP activation. In mACE2 KO mice, NRICM101 could ameliorate the disease progression of acid-induced ALI. In conclusion, the established mouse model provided an effective platform for researchers to investigate pathological mechanisms and develop therapeutic strategies for ALI, including COVID-19-related ALI.

## 1. Introduction

The renin–angiotensin system (RAS) takes a central role in regulating electrolyte balance and blood pressure homeostasis [1,2]. Abnormal activation of the RAS is associated with the pathogenesis of many diseases [2]. RAS is moderated by three key proteases: renin, angiotensin-converting enzyme (ACE), and angiotensin-converting enzyme type II (ACE2). Renin cleaves angiotensinogen to generate angiotensin I (Ang I), and ACE cleaves angiotensin I (Ang I) to generate angiotensin II (Ang II), which is an important regulator of the RAS and exerts biological effects through the specific receptor Ang II receptor type 1 (AT1R) [1]. Thus, ACE is a crucial enzyme in ACE-Ang II/AT1R signaling in the regulation of RAS [3,4]. In 2000, a homolog of ACE was cloned and termed angiotensin converting enzyme II (ACE2) [5,6]. While ACE cleaves Ang I into Ang II, ACE2 removes a single residue from Ang II to generate angiotensin (1-7) (Ang-(1-7)). Studies indicated a counter-regulatory role for Ang-(1-7) via its specific receptor Mas by opposing many AT1R-mediated effects [7,8,9]. ACE2-Ang-(1-7)/Mas signaling has emerged as a potent negative regulator of ACE-Ang II/AT1R in the RAS, playing an opposing role to ACE in various organ systems, including the heart, kidneys, and lungs [2,10,11]. Based on its physiological functions, ACE2 is considered a protector of the RAS [8].

In December 2019, a severe acute respiratory syndrome coronavirus 2 (SARS-CoV-2) and coronavirus disease (COVID-19) outbreak occurred in Wuhan, China [12]. As of March 2023, COVID-19 has infected more than 676 million people and caused the death of over 6 million people worldwide [13]. The COVID-19 pandemic has posed severe threats to populations, social structures, and economic growth [14]. Nevertheless, over the past three years, it is still prudent to accelerate investigations of new therapeutic advances that can effectively treat COVID-19. SARS-CoV-2 enters human cells by the binding of its spike protein to ACE2 on the cell membrane [15]. In addition to the role of ACE2 in viral entry, the interaction between the viral spike protein and ACE2 leads to the development of severe symptoms of COVID-19, such as inflammation and acute lung injury (ALI) [15]. ACE2 depletion by SARS-CoV-2 spike protein binding disrupts the balance between ACE and ACE2 (i.e., ACE/ACE2) and increases the level of Ang II [16]. This could contribute to the development of ALI patients, as well as other complications of COVID-19 [17,18].

ACE2 is a major cell entry receptor for SARS-CoV-2, which suggests that the high expression of ACE2 would cause an increase in cell infection [19]. It is also important for ACE2 to balance the tissue RAS against antioxidation, anti-inflammatory, and antifibrotic responses through ACE2-Ang-(1-7)/Mas signaling [20,21]. This raised the initial dilemma of whether we should increase ACE2 levels in the lungs to inhibit the injury process or reduce tissue ACE2 levels to decrease SARS-CoV-2 entry and replication. This has been termed the double-edged sword of ACE2 [22,23,24]. One of the challenges in studying the double-edged sword of ACE2 related to SARS-CoV-2 pathogenicity is the limited availability of animal models. Another challenge is the highly infective nature of SARS-CoV-2, and handling and experimentation using this virus must be conducted in a biosafety level 3 (BSL-3) facility [25]. It is fortunate that human ACE2 transgenic (hACE2 TG) and murine ACE2 knockout (mACE2 KO) mice have been generated. Both types of mice are reasonable animal and translational models for studying SARS-CoV-2 infection and COVID-19-related ALI pathogenesis [26,27]. Therefore, hACE2 TG and mACE2 KO mice were used in the present study to examine ACE2 and ALI.

The most common and severe complication of COVID-19 is ALI, which can lead to respiratory failure and death [28]. ALI is characterized by the induction of inflammatory cytokines, is associated with COVID-19 severity, and is a crucial cause of death from COVID-19 [29]. Traditional Chinese medicine (TCM) has long been used to modulate immunity and suppress inflammation, particularly in lung diseases [30,31]. Therefore, TCM and Western medicine can be used in combination to treat lung damage caused by SARS-CoV-2, and TCM effectively provides continuous prevention and treatment of COVID-19 [32,33]. In Taiwan, Chingguan Yihau (NRICM101), which is an effective TCM, exerts both antiviral and anti-inflammatory effects against COVID-19 [34]. Tsai et al. [34] pointed out that pharmacological assays demonstrated the effects of NRICM101 on inhibiting the SARS-CoV-2 spike protein/ACE2 interaction. However, no further studies have explored the therapeutic mechanism and effects of TCM on ACE2 regulation.

In this study, ALI was induced in hACE2 TG and mACE2 KO mice by intratracheal instillation of hydrochloric acid (HCl) to explore the relationship between ACE2 and ALI pathogenesis. Intratracheal instillation of HCl could be used to induce ALI in a mouse model [35,36], and acid-induced ALI could contribute to the development of inflammation and disrupt the balance between ACE and ACE2 in a short time [37]. Our results showed that the pathological features of acid-induced ALI were similar to those of COVID-19-related lung damage. In COVID-19-mediated ALI, ACE2 depletion by SARS-CoV-2 spike protein binding disrupts the balance between ACE and ACE2 [16]. In the present study, acid-induced ALI could also contribute to the development of inflammation and disrupt the balance between ACE and ACE2, and then hACE2 TG and mACE2 KO mice were used to study the role of ACE2 in COVID-19-related ALI pathogenesis and the therapeutically applicable research. Moreover, the effects of NRICM101 administration on acid-induced ALI were evaluated in animal models in this study.

## 2. Materials and Methods

### 2.1. Cell Line and Cell Culture

The human lung adenocarcinoma cell line A549 was obtained from the American Type Culture Collection (Manassas, VA, USA). A549 cells were grown in Dulbecco’s modified Eagle’s medium (DMEM) supplemented with 10% fetal bovine serum (FBS) and incubated in a 5% CO_2_ and 37 °C humidified atmosphere. The cells were washed with Dulbecco’s phosphate-buffered saline (DPBS) and suspended with 0.25% trypsin–EDTA solution for subculture and seeding.

### 2.2. Acid and NRICM101 Treatment of Cells

A549 cells were cultured in complete medium for 24 h in the Control group. The cells were treated with 0.5 mg/mL NRICM101 (Sun Ten Pharmaceutical Co., Taipei, Taiwan) for 24 h in the TCM only group. The cells were treated with 0.1 N HCl for 30 min and then placed in culture medium for 24 h in the Acid group. The cells were treated with 0.1 N HCl (pH = 4) for 30 min and then placed in 0.5 mg/mL NRICM101 for 24 h in the NRICM101 group.

### 2.3. Determination of Cell Viability

The 3-(4,5-dimethylthiazol-2-yl)-2,5-diphenyl tetrazolium bromide (MTT) assay is a standard method for determining cell viability. A549 cells were seeded in 96-well plates (4 × 10^4^ cells/well), incubated at 37 °C, and humidified at 5% CO_2_ for 24 h. Following attachment, the cells were treated with different conditions. Subsequently, the medium was removed, and the cells were incubated with 5 mg/mL MTT solution for 4 h. The supernatant in the wells was aspirated, and dimethyl sulfoxide (DMSO) was added to dissolve the purple formazan crystals. Finally, the absorbance of the produced formazan solution was measured at 570 nm (BioTek Instruments, Winooski, VT, USA). The results are expressed as the cell viability (%) relative to the Control group.

### 2.4. Determination of ROS Production

DCFH-DA (Sigma Life Science, St. Louis, MO, USA) is a fluorogenic dye. DCFH-DA penetrated the cell membrane and was hydrolyzed by esterase to form a nonfluorescent compound. DCFH-DA was added to A549 cells and placed in an incubator for 30 min. Then, the cells were washed twice with DPBS, and the cells were resuspended in 0.25% trypsin–EDTA. ROS production in A549 cells was analyzed by a fluorescence spectrophotometer (F-2700; Hitachi High Technologies, Pleasanton, CA, USA) at excitation and emission wavelengths of 488 and 525 nm, respectively.

### 2.5. Animal Model

Three genotypes of mice were used in this study: WT, K18-hACE2 TG (strain: B6. Cg-Tg (K18-ACE2)2Prlmn/J), and mACE2 KO (B6;129S5-Ace2tm1Lex/Mmcd). WT and K18-hACE2 TG mice were purchased from the Jackson Laboratory (Jax, Bar Harbor, ME, USA). The first generation of K18-hACE2 TG mice was bred at Bio-LASCO (Taipei, Taiwan) and identified by DNA genotyping. mACE2 KO mice were obtained from Mutant Mouse Regional Resource Centers (MMRRC; Jax). Twelve-week-old WT, K18-hACE2 TG, and mACE2 KO male mice were divided into the Control, Acid, and Acid+TCM groups (*n* = 5 for each group; and a total of 45 mice were used) in this study (Table 1). All animal experiments obeyed the “Guide for the Care and Use of Laboratory Animals published by National Institutes of Health” (NIH Publication No. 85-23, revised 1996) and were approved by the Animal Welfare Committee of National Chiao Tung University (NCTU-IACUC-110002 is proven from 1 August 2021 to 31 July 2024).

### 2.6. Acid and NRICM101 Treatment of Animals

In the Acid group, the mice were treated with 0.1 N HCl via intratracheal instillation twice on Day 0 and Day 2 (2 μL/g of body weight). In the Acid+TCM group, the mice were treated with acid intratracheal instillation twice on Day 0 and Day 2 and then treated with NRICM101 by oral gavage for 4 days starting from Day 3 to Day 6 (3 g/kg of body weight/day). The Control group was treated with PBS (Gibco-Invitrogen, Paisley, UK) via intratracheal instillation. The experiment was divided into 3 groups (*n* = 5), all mice were sacrificed 4 days after intratracheal instillation and physiopathological examinations of the lungs were performed. TST and FST were used to evaluate animal activity. Body weight, TST, and FST were measured daily before the animals were sacrificed.

### 2.7. Lung Histopathology

Lung tissues were isolated from WT, K18-hACE2, and mACE2 KO mice after different conditions of treatment. After collection of tissue samples, lungs were soaked in 10% formaldehyde overnight, embedded in paraffin, and cut into 6 μm thick sections on acid-pretreated slides for hematoxylin and eosin (H&E) and Masson’s trichrome staining. The stained pathological sections were photographed and morphometric analysis by a computerized microscope equipped with a high-resolution video camera (BX 51; Olympus, Tokyo, Japan). The degree of lung injury was scored by lung injury scoring system [38]. The criteria of Ashcroft’s method were applied to score pulmonary fibrosis [39].

### 2.8. Enzyme-Linked Immunosorbent Assay (ELISA)

Lung IL-6 and TNF-α levels were analyzed by sandwich ELISA kits (Abcam, Cambridge, MA, USA). Homogenates of mouse lung tissue were incubated in 96-well ELISA plates with primary antibodies. After the addition of biotinylated capture antibodies, the plates were washed and reacted with HRP-conjugated streptavidin. Tetramethylbenzidine (TMB) was used to detect the level of IL-6 and TNF-α, and the results were measured at 450 nm using a microplate reader (Thermo Scientific, Waltham, MA, USA).

### 2.9. Western Blotting

Quantified mice lung homogenates were set as equivalent protein content of 25 μg, separated on 12% sodium dodecyl sulfate–polyacrylamide gel electrophoresis (SDS–PAGE), and then transferred to polyvinylidene fluoride membranes (Immo-bilon-PTM; Millipore, Bedford, MA, USA). Primary antibodies against ACE were purchased from Genetex (Irvine, CA, USA); STAT3, phospho-STAT3, ERK1/2, phospho-ERK1/2, and β-actin were obtained from Cell Signaling Technology (Beverly, MA, USA).

Visualized for protein analysis was detected by enhanced chemiluminescence (Immobilon Western Chemiluminescent HRP Substrate; Millipore, Billerica, MA, USA). PAGE membranes were exposed in Lumi-Film Chemiluminescent Detection Film (Roche, Indianapolis, IN, USA). Scion Image software (Scion, Frederick, MD, USA) was used to analyze the data from the densitometric quantification based on the ratio to the endogenous control β-actin.

### 2.10. Gelatin Zymography

Gelatin zymography analysis was used to determine the activity of MMP-2 and MMP-9 in lung tissue. The detailed experimental procedure of gelatin zymography analysis was described in our previous study [40].

### 2.11. Statistical Analysis

All data and figures were expressed as the mean ± standard deviation (SD). Differences between each two experimental groups were performed by Student’s *t*-test. One-way analysis of variance (ANOVA) was performed to evaluate differences among multiple groups.

## 3. Results

### 3.1. NRICM101 Promoted Viability and Reduced Oxidative Stress in Acid-Treated A549 Cells

To investigate the potential of the TCM NRICM101 to alleviate cytotoxicity and oxidative stress resulting from acid treatment in the lung cell line A549, we subjected the cells to acid (0.1 N HCl; pH = 4) for 30 min, followed by the administration of NRICM101 for 24 h. Subsequently, cell viability in the TCM only (i.e., only treatment by NRICM101), Acid, and Acid+TCM groups were 97, 68, and 87%, respectively, compared to the Control group (Figure 1a). To assess the level of intracellular reactive oxygen species (ROS) following acid and NRICM101 treatment, 2,7-dichlorofluorescin (DCF)-derived fluorescence was used to analyze cellular ROS levels. The DCF-derived fluorescence in the TCM only, Acid and Acid+TCM groups was 1.5-, 6.0-, and 4.2-fold higher, respectively, than that in the Control group (Figure 1b,c).

### 3.2. ACE2 and NRICM101 Enhanced Vitality in Mice with Acid-Induced ALI

C57BL/6 mice (wild type, WT), K18-hACE2 transgenic mice (K18-hACE2 TG), and mACE2 knockout mice (mACE2 KO) were treated with PBS by intratracheal instillation twice on Day 0 and Day 2 in the Control group. The mice were treated with acid intratracheal instillation twice on Day 0 and Day 2 in the Acid group. The mice were treated with acid intratracheal instillation twice on Day 0 and Day 2 and then administered NRICM101 for 4 days from Day 3 to Day 6 in the Acid+TCM group (Table 1). The health, body weight, and vitality of the animals were monitored daily until sacrifice. Body weight was markedly decreased throughout the entire period of acid intratracheal instillation and slightly recovered after the cessation of acid instillation. In the acid group, the body weight loss of WT mice, K18-hACE2 TG, and mACE2 KO are 10%, 5%, and 20%, respectively. Notably, mACE2 KO mice with acid intratracheal instillation exhibited rapid and sustained declines in body weight compared with the other mice. However, the administration of NRICM101 did not significantly affect the body weight of the mice (Figure 2a).

The effects of ACE2 and NRICM101 on acid-induced ALI mice were examined by the tail suspension test (TST) and forced swimming test (FST), and the results are shown in Figure 2b,c. In the Acid group, the TST results of WT, K18-hACE2 TG, and mACE2 KO mice were significantly reduced by approximately 76, 86, and 67%, respectively, and the FST results of WT, K18-hACE2 TG, and mACE2 KO mice were significantly reduced by approximately 75, 90, and 62%, respectively, compared with those in the Control group. In the Acid+TCM group, the TST and FST results of WT, K18-hACE2 TG, and mACE2 KO mice were increased by approximately 86, 89, and 83% and 89, 97, and 86% compared with those in the Control group, respectively. The acid-induced ALI mice that were administered NRICM101 exhibited effectively increased vitality.

### 3.3. ACE2 and NRICM101 Reversed Histopathological Changes in ALI Mice

Analysis of hematoxylin and eosin (H&E)-stained alveolar and bronchiolar sections from WT, K18-hACE2 TG, and mACE2 KO mice showed a progressive inflammatory process (Figure 3a). White blood cells (WBCs) infiltrated a greater area of the lung into adjacent alveolar spaces with airway epithelial thickening by acid instillation. In the Acid group, the lung injury score of WT, K18-hACE2 TG, and mACE2 KO mice was increased by approximately 3.0-, 2.1-, and 3.8-fold, respectively, compared with that in the Control group (Figure 3b). The extent of infiltration and airway epithelial thickening was decreased with TCM treatment. In the Acid+TCM group, the lung injury score of WT, K18-hACE2 TG, and mACE2 KO mice was approximately 1.7-, 1.2-, and 3.0-fold, respectively, compared with that in the Control group (Figure 3b).

The lung tissue sections were also stained with Masson’s trichrome to examine fibrotic pathology. Massive collagen deposition was observed throughout the alveolar and bronchial walls by acid instillation. In the Acid group, the Ashcroft score of WT, K18-hACE2 TG, and mACE2 KO mice was approximately 4.8-, 3.4-, and 6.1-fold, respectively, compared with that in the Control group (Figure 3c). The collagen deposition was decreased with TCM treatment. In the Acid+TCM group, the Ashcroft score of WT, K18-hACE2 TG, and mACE2 KO mice was approximately 3.2-, 2.2-, and 3.6-fold, respectively, compared with that in the Control group (Figure 3c). Moreover, acid-induced ALI pathological changes in mACE2 KO mice were far greater than those in K18-hACE2 TG mice. The pathological features of ALI mice, including WBC infiltration, alveolar disruption, and thickened airway epithelium by acid instillation, were slightly to moderately restored by NRICM101 administration.

### 3.4. ACE2 and NRICM101 Ameliorated Pulmonary Inflammation in ALI Mice

In the Acid group, IL-6 and TNF-α levels in the bronchoalveolar lavage fluid (BALF) were increased by 2.0- and 1.6-fold in WT mice, 1.1- and 1.1-fold in K18-hACE2 TG mice and 2.5- and 2.2-fold in mACE2 KO mice, respectively, compared with those in the Control group (Figure 4). In particular, IL-6 levels in mACE2 KO mice were significantly higher than those in the other mice with acid intratracheal instillation. In the Acid+TCM group, IL-6 and TNF-α levels in BALF were reduced by approximately 1.3- and 1.4-fold in the WT mice; 1.0- and 1.0-fold in K18-hACE2 TG mice; and 1.8- and 1.5-fold in mACE2 KO mice, respectively, compared with those in the Control group (Figure 4). Interestingly, levels of inflammatory cytokines in K18-hACE2 TG mice with acid instillation and NRICM101 administration were similar to their basal levels.

### 3.5. NRICM101 Administration Countered the Imbalance in ACE in ALI Mice

The protein expression of ACE was investigated to elaborate on the RAS mediators associated with acid-induced ALI. In the Acid group, ACE expression in WT, K18-hACE2 TG, and mACE2 KO mice was increased by approximately 2.0-, 1.2-, and 6-fold, respectively, compared with that in the Control group. There was no change in ACE expression in K18-hACE2 TG mice with acid intratracheal instillation. In the Acid+TCM group, ACE expression in WT and mACE2 KO mice was decreased by 75% and 70%, respectively, compared with that in the Acid group (Figure 5).

### 3.6. ACE2 and NRICM101 Alleviated STAT3 and ERK1/2 Phosphorylation in the Lungs of ALI Mice

This present study indicated that an increased level of inflammatory cytokines was observed in mice after acid intratracheal instillation. IL-6 and TNF-α participated in inflammatory responses by activating the STAT3 and ERK1/2 signaling pathways. Therefore, pERK1/2 and p STAT3 in lung tissues were detected by Western blotting (Figure 6a).

The results showed that pulmonary phosphorylated STAT3 (p-STAT3) levels under acid instillation in WT and mACE2 KO mice were significantly increased by 3.1- and 4.8-fold, respectively, but there was no significant effect on K18-hACE2 TG mice (Figure 6b). Similarly, pulmonary phosphorylated ERK1/2 (p-ERK1/2) levels under acid instillation in WT and mACE2 KO mice were significantly increased by 3.3- and 4.2-fold, respectively (Figure 6c). Conversely, WT and mACE2 KO mice in the Acid+TCM group had a significantly downregulated expression of p-STAT3 and p-ERK1/2 compared to the Acid group (Figure 6).

### 3.7. ACE2 and NRICM101 Modulated Pulmonary Gelatinase Activity in Mice with Acid-Induced ALI

Gelatinase containing MMP-2 and MMP-9 was measured by gelatin zymography (Figure 7a). In the Acid group, the MMP-2 activity of WT, K18-hACE2 TG, and mACE2 KO mice was increased by approximately 1.9-, 1.1-, and 2.7-fold, respectively, compared with that in the Control group (Figure 7b). Similarly, MMP-9 activity in WT, K18-hACE2 TG, and mACE2 KO mice in the Acid group was increased by approximately 2.6-, 1.3-, and 3.7-fold, respectively, compared to the Control group (Figure 7c). Gelatinase activity was significantly increased in the lungs of WT and mACE2 KO mice after acid instillation, but there was no significant change in the lungs of K18-hACE2 TG mice.

Furthermore, in the Acid+TCM group, the MMP-2 activity of WT, K18-hACE2 TG, and mACE2 KO mice was reduced by 0.9-, 1.0-, and 2.1-fold, respectively, compared to the Control group. However, MMP-9 activity was reduced 2.0-, 1.3-, and 3.0-fold in WT, K18-hACE2 TG, and mACE2 KO mice, respectively. The gelatinase activity of the WT and mACE2 KO mice in the Acid+TCM group was significantly changed compared to that in the Acid group.

## 4. Discussion

Recently, the animal model of K18-hACE2 TG mice has been widely used in COVID-19 studies, including studies of vaccine evaluation and antiviral agents to combat SARS-CoV-2 [41,42]. The rapid inflammatory response and observed respiratory pathology after K18-hACE2 TG mice were challenged with SARS-CoV-2 resembles COVID-19 [42]. In fact, K18-hACE2 TG mice were originally and successfully developed for a SARS-CoV study in 2007 [43]. ACE2 has physiological roles as a SARS-CoV-2 entry receptor and in acute respiratory distress syndrome (ARDS) or ALI [44]. Even though human ACE2 (hACE2), which is the entry receptor for SARS-CoV-2, has been well established, the contribution of the enzymatic functions of ACE2 to the pathogenesis of COVID-19-related lung injury has been a matter of debate [45]. Therefore, this study aims to establish an optimal ACE2 modification mice model that can reproduce multiple characteristics of ALI to study the role of ACE2 and look for therapeutic applications. Recently, ROS generation has been suggested as a vital factor in COVID-19 and causes many disturbing effects on the body [46,47]. In this study, the mice with genetic modifications absence of ACE2 (mACE2 KO) is presented as ACE2 depletion by SARS-CoV-2 spike protein binding, and acid-induced ALI could contribute to the development of oxidative stress and inflammation and disrupts the balance between ACE and ACE2 [37]. Thus, acid-induced ALI in ACE2 modifications mice were used as an alternative to COVID-19-mediated ALI under the safety condition.

In this study, acid intratracheal instillation in mice induced an inflammatory response and induced the production of the inflammatory cytokines IL-6 and TNF-α in the lungs. Several pulmonary diseases, such as ALI, ARDS, asthma, COPD, and COVID-19 pneumonia, are associated with abnormal IL-6 and TNF-α expression [48,49,50]. Our results showed that IL-6 and TNF-α levels in the lungs of WT and mACE2 KO mice were significantly increased 4 days after acid instillation. Moreover, the lung sections of WT and ACE2 KO mice showed serious immunocyte infiltration and alveolar injury. These pathological features were also observed in the lungs of K18-hACE2 TG mice with mild symptoms. These data suggest that the overproduction of ACE2 and induction of the ACE2-Ang-(1-7)/Mas axis could reduce the inflammation induced by acid treatment. This finding was confirmed by many publications [51,52,53]. Activation of the ACE2-Ang-(1-7)/Mas receptor axis is absent in mACE2 KO mice, which exhibited severe injuries and imbalance in the RAS system compared with K18-hACE2 TG mice [37]. Additionally, an increase in ACE is associated with ALI. Based on these findings, robust increases in ACE protein expression were measured in acid-induced ALI. The upregulation of ACE and depletion of ACE2 could result in an imbalance in ACE/ACE2 [40,54,55], such as increasing the ACE/ACE2 ratio to accelerate the disease progression of acid-induced ALI or COVID-19 ALI.

ALI is characterized by the induction of inflammatory cytokines, such as IL-6 and TNF-α. These cytokines could participate in the MAPK, ERK1/2, and/or STAT3 pathways [56] and activate extracellular matrix production, such as collagen and gelatin [57]. In addition, MMPs have been observed in a wide variety of pulmonary pathologies and could be regulated by MAPK signaling [58]. Members of the MMP family, especially the gelatinases MMP-2 and MMP-9, have been implicated in pulmonary inflammation and asthma, ALI, and pulmonary fibrosis [58,59]. Some studies have shown that ACE2 mediated the MMP-2 and MMP-9 expression [60,61]. We proposed that MMP regulation was responsible for the anti-inflammatory and/or antifibrotic effects on the lungs during acid-induced injury. MMP-2, which is synthesized by structural cells, including endothelial cells, epithelial cells, and fibroblasts, is associated with impaired tissue remodeling, thus leading to pathological collagen deposition and pulmonary fibrosis [62,63]. In contrast, MMP-9 is mainly secreted by neutrophils, eosinophils, mast cells, and alveolar macrophages and is associated with acute inflammation [64]. Studies have shown that SARS-CoV-2 infection can upregulate MMP-2 or MMP-9 expression in lung tissue [65]. Additionally, there is evidence to suggest that ACE2 deficiency can contribute to the dysregulation of gelatinase activity in COVID-19 [66,67]. Our data are consistent with the conclusions of previous studies. The present study showed that ACE2 reduced oxidative stress, decreased lung injury, ameliorated pulmonary inflammation, alleviated STAT3 and ERK1/2 phosphorylation, and finally counteracted the pathogenic process of ALI. ACE2 played a role in acid-mediated ALI which would be a therapeutically target for further research.

There have been numerous reports on the antiviral properties of dozens of Chinese herbs and hundreds of TCM ingredients [68]. TCM formulations have also been used to treat previous pandemics, including severe acute respiratory syndrome coronavirus (SARS-CoV) and Middle East respiratory syndrome coronavirus (MERS-CoV) outbreaks [69,70]. In recent years, TCM formulations have been developed to treat COVID-19 [71,72]. In Taiwan, the prescription used in general mild patients is Taiwan NRICM101, which was announced in May 2020, and the clinical results were reported in August 2022 [73]. In a mouse model, Taiwan NRICM101 exerted antiviral effects by reducing the affinity between the viral spike protein and ACE2 and suppressed IL-6 and TNF-α production in alveoli [34]. Thus, evidence can support the therapeutic effects of NRICM101 on COVID-19 treatment with at least two targeting mechanisms: preventing SARS-CoV-2 infection and reducing the inflammation induced by the virus. According to Tsai et al. [34], NRICM101 has a multitarget mechanism of action. NRICM101 can block the binding of the ACE2 protein and spike protein to prevent virus infection and prevent viral replication by inhibiting the activity of viral 3CLpro [34]. Our data show that NRICM101 is effective against COVID-19, which corresponds to the more recent publications by Chang et al. [74] and Singh et al. [75].

## 5. Conclusions

The COVID-19 pandemic has caused continuing challenges to global health. It is still prudent to accelerate investigations of new therapeutic advances that can effectively treat COVID-19. Effective animal models for COVID-19 studies need to be developed because using SARS-CoV-2 greatly limits routine research. The most common and severe complication of COVID-19 is ALI, which can lead to respiratory failure and death. ACE2 is the receptor for SARS-CoV-2 entry into cells, and ACE2 plays a protective role in diseases of the respiratory system, such as ALI. It is important to validate the interaction between ACE2 and ALI, especially COVID-19-related ALI. The present study is the first report that used mice with *Ace2* gene modifications, including hACE2 transgenic mice and ACE2 knockout mice, to establish mouse models of acid-induced ALI and study the role of ACE2 in molecular pathogenesis. Our results indicated that ACE2 could counteract the proinflammatory effect of ALI and attenuate ALI pathogenesis. ACE2 effectively alleviates the pathogenic process of ALI by regulating MAPK, STAT3, and MMP activation. Finally, the TCM formula NRICM101, which has been used to prevent and treat COVID-19, shows potential for treating ALI.

## Figures and Tables

**Figure 1 medicina-59-01554-f001:**
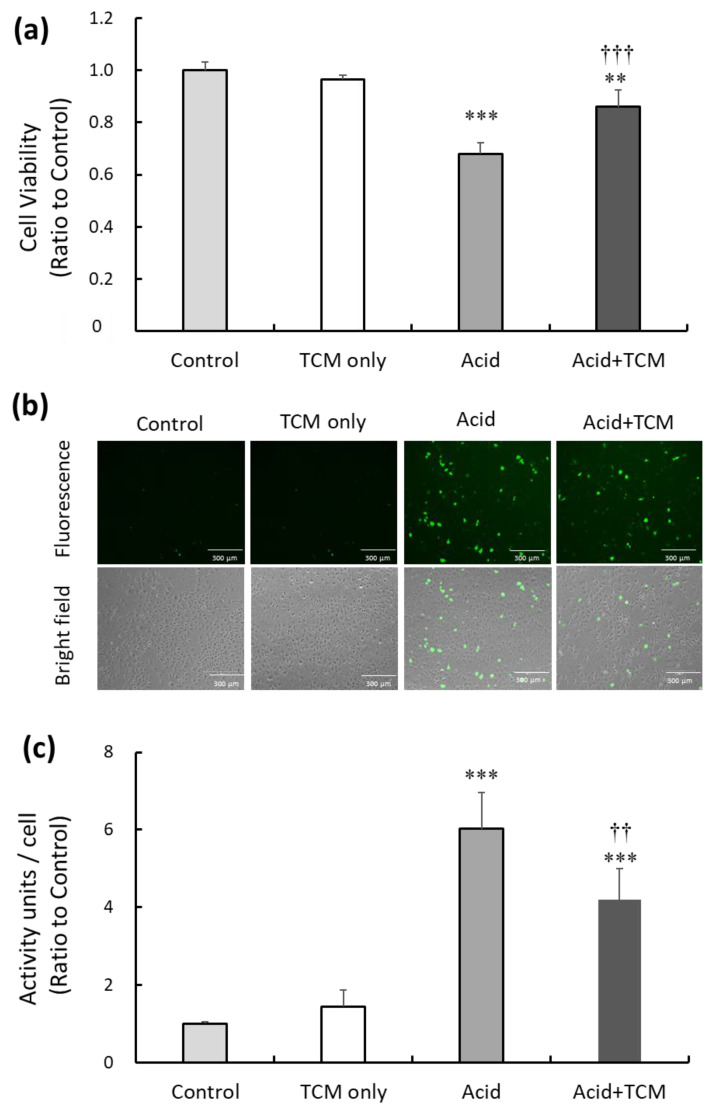
Effect of TCM and NRICM101 treatment on cell viability and ROS production in A549 cells exposed to acidic medium. (**a**) Cell viability in the Control, TCM only, Acid, and Acid+TCM groups was detected by MTT assays. Cell viability in the nonacid and non-NRICM101 treatment groups was calculated as 100% and defined as the control value. The cells were treated with medium containing HCl (pH = 4) for 30 min in Acid group, and then treated with 0.5 mg/mL NRICM101 for 24 h in Acid+TCM group. (**b**) The level of ROS in A549 cells was measured by DCF fluorescence, and photographs of 2,7-dichlorofluorescin diacetate (DCFH-DA)-stained A549 cells are shown. (**c**) Histogram values reflect the mean ± SD (*n* = 5 for each treatment), and the values in the plots are the relative DCFH-DA-derived activity units/cell fluorescence relative to the Control group, which was 1. ** *p* < 0.01 and *** *p* < 0.001 vs. Control group; †† *p* < 0.01 and ††† *p* < 0.001 vs. Acid group. Control group: A549 cells cultured in complete medium for 24 h. TCM only group: The cells were treated with 0.5 mg/mL NRICM101 for 24 h. Acid group: The cells were treated with 0.1 N HCl (pH = 4) for 30 min. Acid+TCM group: The cells were treated with 0.1 N HCl for 30 min and then placed in medium containing 0.5 mg/mL NRICM101 for 24 h.

**Figure 2 medicina-59-01554-f002:**
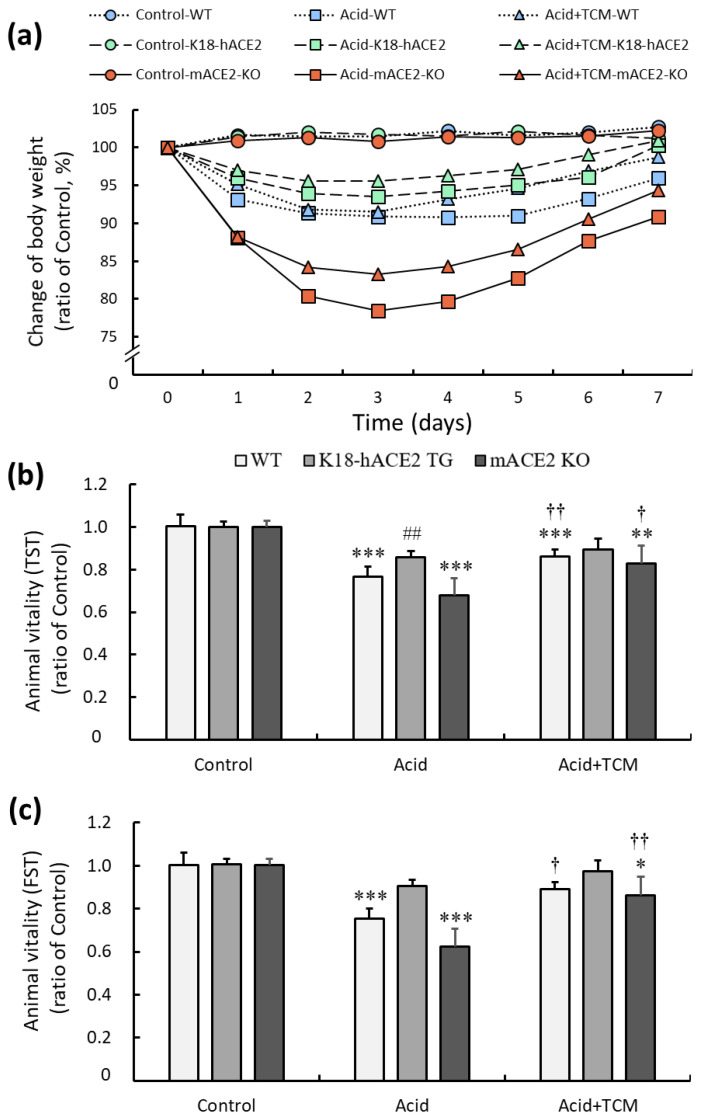
Physiological changes in acid-induced ALI mice administered NRICM101. (**a**) Changes in the body weights of the mice during the experiments. The average body weight on Day 0 in each group was calculated as 100%. (**b**) The changes in animal vitality, as detected by the TST, were determined and compared to the average TST result on Day 0 in each group, which was calculated as 100%. (**c**) The changes in animal vitality, as detected by FST, were determined and compared to the average FST result on Day 0 in each group, which was calculated as 100%. All values are expressed as the mean ± SD for each group (*n* = 5); * *p* < 0.05, ** *p* < 0.01 and *** *p* < 0.001 vs. Control group; † *p* < 0.05 and †† *p* < 0.01 vs. Acid group; ## *p* < 0.01 vs. WT mice. TST: tail suspension test; FST: forced swimming test. Control group: The mice were treated twice with phosphate-buffered saline (PBS) intratracheal instillation. Acid group: The mice were treated twice with acid intratracheal instillation. Acid+TCM group: The mice were treated twice with acid intratracheal instillation and then administered NRICM101 for 4 days.

**Figure 3 medicina-59-01554-f003:**
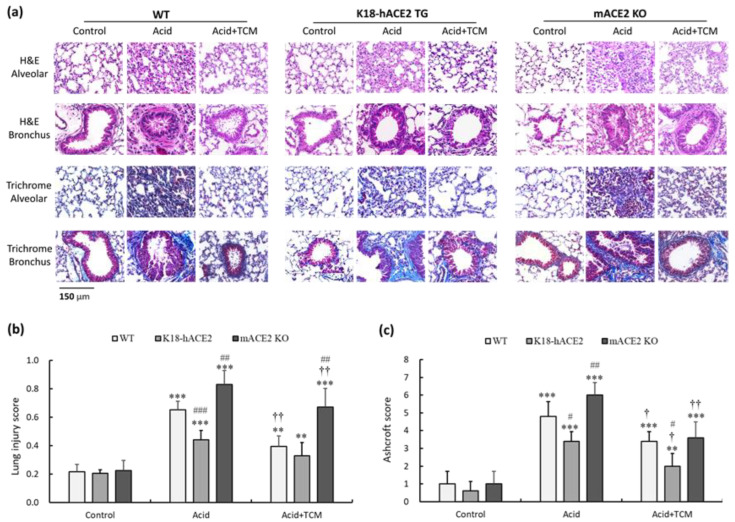
Bronchial and alveolar pathologic findings of acid-induced ALI mice and ALI mice administered NRICM101. (**a**) Bronchial and alveolar sections of mouse lungs were stained with H&E and Masson’s trichrome to determine their pathological features. (**b**) Lung injury score was assessed by histological scores. (**c**) Ashcroft score was performed for lung fibrosis quantification. All values are expressed as the mean ± SD in each group (*n* = 5). ** *p* < 0.01 and *** *p* < 0.001 vs. Control group; † *p* < 0.05 and †† *p* < 0.001 vs. Acid group; # *p* < 0.05, ## *p* < 0.01 and ### *p* < 0.001 vs. WT mice.

**Figure 4 medicina-59-01554-f004:**
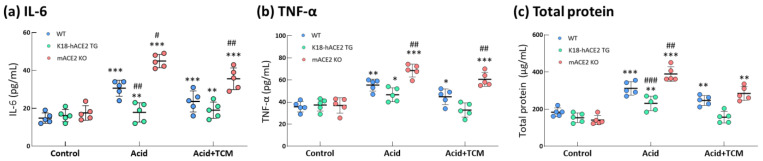
Changes in total protein and cytokines in the lungs of ALI mice administered NRICM101. Before the animals were sacrificed, BALF was collected from the mice to examine the IL-6 (**a**), TNF-α (**b**), and total protein (**c**) levels. The results showed that BALF total protein and cytokine levels in acid-treated mice were alleviated by NRICM101 administration. All values are expressed as the mean ± SD in each group (*n* = 5). * *p* < 0.05, ** *p* < 0.01 and *** *p* < 0.001 vs. Control group; # *p* < 0.05, ## *p* < 0.01, ### *p* < 0.001 vs. WT mice.

**Figure 5 medicina-59-01554-f005:**
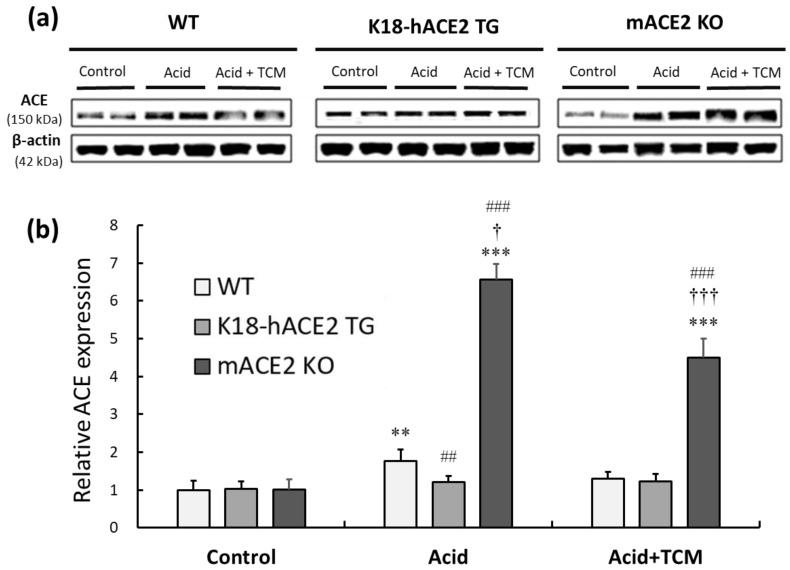
Relative ACE expression in the lungs of ALI mice administered NRICM101. (**a**) ACE expression (approximately 150 kDa) was validated by Western blotting. (**b**) The relative ACE expression level (ACE/β-actin) was quantitated. Acid intratracheal instillation significantly induced pulmonary ACE expression, and the level of ACE was decreased in mice administered NRICM101. All values are expressed as the mean ± SD in each group (*n* = 5). ** *p* < 0.01 and *** *p* < 0.001 vs. Control group; † *p* < 0.05 and ††† *p* < 0.001 vs. Acid group; ## *p* < 0.01 and ### *p* < 0.001 vs. WT mice.

**Figure 6 medicina-59-01554-f006:**
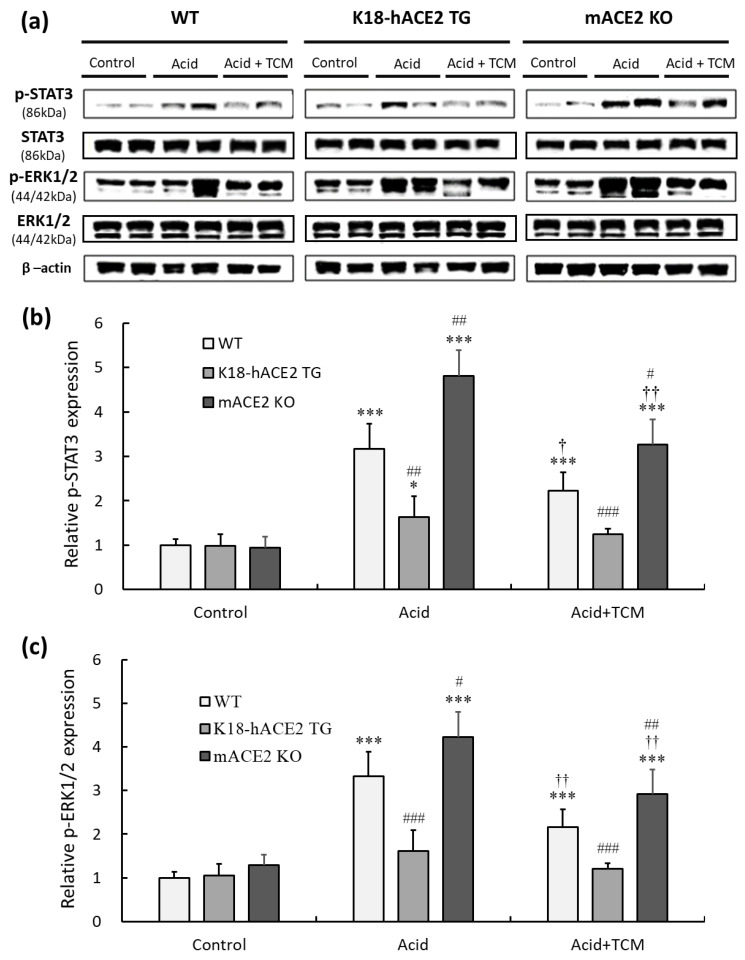
Relative p-STAT3 and p-ERK 1/2 expression in the lungs of ALI mice administered NRICM101. (**a**) p-STAT3 (150 kDa) and p-ERK1/2 (44/42 kDa) expression were validated by Western blotting. (**b**) Relative quantitative analysis of p-STAT3. (**c**) Relative quantitative analysis of p-ERK1/2. Acid intratracheal instillation significantly induced pulmonary p-STAT3 and p-ERK 1/2 expression, and the increases in p-STAT3 and p-ERK 1/2 were ameliorated in the mice administered NRICM101. All values are expressed as the mean ± SD in each group (*n* = 5); * *p* < 0.05 and *** *p* < 0.001 vs. Control group; † *p* < 0.05 and †† *p* < 0.01 vs. Acid group; # *p* < 0.05, ## *p* < 0.01, ### *p* < 0.001 vs. WT mice.

**Figure 7 medicina-59-01554-f007:**
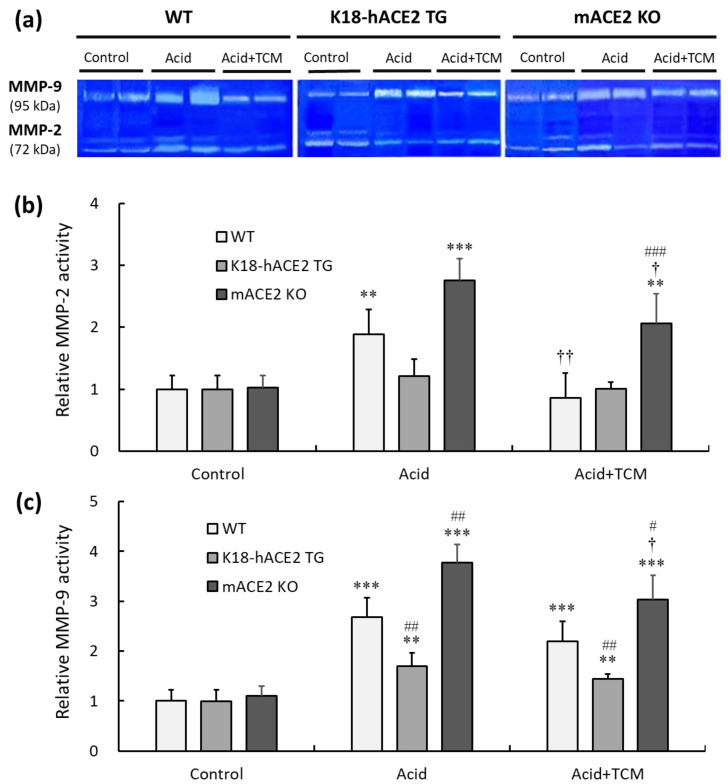
Relative MMP-2 and MMP-9 activity in the lungs of ALI mice administered NRICM101. (**a**) MMP-2 (72 kDa) and MMP-9 (95 kDa) activity in the lung was detected by gelatin zymography. (**b**) Relative quantitative analysis of MMP-2 activity. (**c**) Relative quantitative analysis of MMP-9 activity. Acid intratracheal instillation significantly induced pulmonary MMP-2 and MMP-9 activity, and the increase in MMP activity was alleviated in mice administered NRICM101. All values are expressed as the mean ± SD in each group (*n* = 5). ** *p* < 0.01 and *** *p* < 0.001 vs. Control group; † *p* < 0.05 and †† *p* < 0.01 vs. Acid group; # *p* < 0.05, ## *p* < 0.01, ### *p* < 0.001 vs. WT mice.

**Table 1 medicina-59-01554-t001:** Groups of in vivo experimental mice.

Mice	Group	*n*	PBS	Acid	NRICM101
WT	Control	5	+	−	−
	Acid	5	−	+	−
	Acid+TCM	5	−	+	+
K18-hACE2 TG	Control	5	+	−	−
	Acid	5	−	+	−
	Acid+TCM	5	−	+	+
mACE2 KO	Control	5	+	−	−
	Acid	5	−	+	−
	Acid+TCM	5	−	+	+

WT, K18-hACE2 TG, and mACE2 KO mice were treated twice with PBS intratracheal instillation (Control group). The mice were treated twice with acid intratracheal instillation (Acid group). The mice were treated twice with acid intratracheal instillation and then administered NRICM101 for 4 days (Acid+TCM group). For acid intratracheal instillation, 0.1 N HCl was intratracheally instillated once every three days, twice in total (2 μL/g of body weight). For NRICM101 administration, NRICM101 was administered consecutively for 4 days (3 g/kg of body weight/day).

## Data Availability

The data presented in this study are available upon request from the corresponding author.

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
