# Peer review of "ACE2 and a Traditional Chinese Medicine Formula NRICM101 Could Alleviate the Inflammation and Pathogenic Process of Acute Lung Injury"

_medicina, 2023, doi:10.3390/medicina59091554_

Round 1
Reviewer 1 Report
In this manuscript Lin et al have investigated that ACE2 and a Traditional Chinese Medicine Formula NRICM101 Could Alleviate the Inflammation and Pathogenic Process of Acute Lung Injury. The study is good with few concerns that need to be addressed in the manuscript.
1. How can author justify that acid-induced ALI is really a good model to understand COVID-19 mediated-ALI since ACE2 is required for Corona virus-mediated ALI. In this study ACE2 is shown to reduce ALI induced by acid.
2. Use of mice model is appropriate to understand coronavirus-mediated ALI but acid behaves very differently as it is a chemical compound and has lots of other effects via its ability to bind many receptors or proteins inside the cell. Because of this we see a lot of ROS generation. Relating its effect with a virus induced ALI is not justifiable. Author could reframe their claim in the study indicating the role of only acid-induced ALI which is also a big concern.
3. It is good to see that ACE2 has a role in Acid-mediated ALI which would be therapeutically target to address this model of human pathology. This should be highlighted in the dicussion.
English language is good.
Author Response
Response to Reviewer 1 Comments
Responses to Reviewer’s comments:
We deeply appreciate the reviewer’s time and efforts to review the manuscript, “ACE2 and a Traditional Chinese Medicine Formula NRICM101 Could Alleviate the Inflammation and Pathogenic Process of Acute Lung Injury” (Medicina-2520593)”, and provide very critical and insightful comments. The comments raised were extremely helpful and have been fully integrated into the revised submission. We sincerely appreciate you for the detailed comments and suggestions to improve the value of manuscript. Each of these comments have been addressed.
The revised portions were highlighted in red in the revised manuscript. The followings are our point-by-point responses to the comments:
Point 1: How can author justify that acid-induced ALI is really a good model to understand COVID-19 mediated-ALI since ACE2 is required for Corona virus-mediated ALI. In this study ACE2 is shown to reduce ALI induced by acid.
Response 1: Thanks for your precious suggestion. In COVID-19 mediated-ALI, ACE2 depletion by SARS-CoV-2 spike protein binding disrupts the balance between ACE and ACE2 [Banu et al.,2020]. Acid-induced ALI could contribute to the development of inflammation and disrupts the balance between ACE and ACE2 in a short time [Imai et al., 2005]. In this study, the overproduction of ACE2 and induction of the ACE2-Ang-(1-7)/Mas axis could reduce the inflammation induced by acid treatment. Activation of the ACE2-Ang-(1-7)/Mas receptor axis is absent in mACE2 KO mice, which exhibited severe injuries and imbalance in the RAS system compared with K18-hACE2 TG mice. Therefore, acid-induced ALI in ACE2 genomic modifications mice were used as an optimal animal model to evaluate the therapeutically appliable research.
According to the comment, we revised the statements as followings:
“In this study, ALI was induced in hACE2 TG and mACE2 KO mice by intratracheal instillation of hydrochloric acid (HCl) to explore the relationship between ACE2 and ALI pathogenesis. Intratracheal instillation of HCl could be used to induce ALI in a mouse model [Basoalto et al, 2021; Tavares et al, 2019], and acid-induced ALI could contribute to the development of inflammation and disrupts the balance between ACE and ACE2 in a short time [Imai et al., 2005]. Our results showed that the pathological features of acid-induced ALI were similar to those of COVID-19-related lung damage. In COVID-19 mediated-ALI, ACE2 depletion by SARS-CoV-2 spike protein binding disrupts the balance between ACE and ACE2 [Banu et al.,2020]. In this study, acid-induced ALI could also contribute to the development of inflammation and disrupts the balance between ACE and ACE2, and then hACE2 TG and mACE2 KO mice were used to study the role of ACE2 in COVID-19-related ALI pathogenesis and the therapeutically appliable research.” (Lines 91-102 in the Introduction section of revised manuscript)
Basoalto, R.; Damiani, L.F.; Bachmann, M.C.; Fonseca, M.; Barros, M.; Soto, D.; Araos, J.; Jalil, Y.; Dubo, S.; Retamal, J.; et al. Acute lung injury secondary to hydrochloric acid instillation induces small airway hyperresponsiveness. Am J Transl Res 2021, 13, 12734-12741.
Tavares, A.H.; Colby, J.K.; Levy, B.D.; Abdulnour, R.E. A Model of Self-limited Acute Lung Injury by Unilateral Intra-bronchial Acid Instillation. J Vis Exp 2019, doi:10.3791/60024.
Banu, N.; Panikar, S.S.; Leal, L.R.; Leal, A.R. Protective role of ACE2 and its downregulation in SARS-CoV-2 infection leading to Macrophage Activation Syndrome: Therapeutic implications. Life Sci 2020, 256, 117905, doi:10.1016/j.lfs.2020.117905.
Imai Y, Kuba K, Rao S, Huan Y, Guo F, Guan B, Yang P, Sarao R, Wada T, Leong-Poi H, Crackower MA, Fukamizu A, Hui CC, Hein L, Uhlig S, Slutsky AS, Jiang C, Penninger JM. Angiotensin-converting enzyme 2 protects from severe acute lung failure. Nature. 2005 Jul 7;436(7047):112-6. doi: 10.1038/nature03712. PMID: 16001071; PMCID: PMC7094998.
Point 2: Use of mice model is appropriate to understand coronavirus-mediated ALI but acid behaves very differently as it is a chemical compound and has lots of other effects via its ability to bind many receptors or proteins inside the cell. Because of this we see a lot of ROS generation. Relating its effect with a virus induced ALI is not justifiable. Author could reframe their claim in the study indicating the role of only acid-induced ALI which is also a big concern.
Response 2: Thank you very much for the affirmation of this research article. The lung injury induced by SARS-CoV-2 is more complex than acid-induced exactly. However, the challenge in studying SARS-CoV-2 pathogenicity is the limited availability of animal models. Therefore, this study aims to establish an animal model that can reproduce multiple characteristics of ALI to study therapeutic applications. Recently, ROS generation has been suggested as a crucial factor in COVID-19 and has many disturbing effects on the body [Cecchini R, 2020; Delgado-Roche L, 2020]. SARS-CoV-2 enters human cells by binding its spike protein to the host cell membrane ACE2. ACE2 depletion by SARS-CoV-2 spike protein binding disrupts the balance between ACE and ACE2 and contributes to ALI's development. In this study, the mice with genetic modifications absence of ACE2 (mACE2 KO) is presented as ACE2 depletion by SARS-CoV-2 spike protein binding, and acid-induced ALI could contribute to the development of inflammation and disrupts the balance between ACE and ACE2 [Imai et al., 2005]. Thus, acid-induced ALI in ACE2 modifications mice were used as an alternative to COVID-19 mediated-ALI under the safety condition.
According to the suggestion, we revised the statements as followings:
“Therefore, this study aims to establish an optimal ACE2 modification mice model that can reproduce multiple characteristics of ALI to study the role of ACE2 and look for therapeutic applications. Recently, ROS generation has been suggested as a crucial factor in COVID-19 and has many disturbing effects on the body [Cecchini R, 2020; Delgado-Roche L, 2020]. In this study, the mice with genetic modifications absence of ACE2 (mACE2 KO) is presented as ACE2 depletion by SARS-CoV-2 spike protein binding, and acid-induced ALI could contribute to the development of oxidative stress, inflammation and disrupts the balance between ACE and ACE2 in a short time [Imai et al., 2005]. Thus, acid-induced ALI in ACE2 modifications mice were used as an alternative to COVID-19 mediated-ALI under the safety condition.” (Lines 415-423 in the Discussion section of revised manuscript)
Imai Y, Kuba K, Rao S, Huan Y, Guo F, Guan B, Yang P, Sarao R, Wada T, Leong-Poi H, Crackower MA, Fukamizu A, Hui CC, Hein L, Uhlig S, Slutsky AS, Jiang C, Penninger JM. Angiotensin-converting enzyme 2 protects from severe acute lung failure. Nature. 2005 Jul 7;436(7047):112-6. doi: 10.1038/nature03712. PMID: 16001071; PMCID: PMC7094998.
Cecchini R, Cecchini AL. SARS-CoV-2 infection pathogenesis is related to oxidative stress as a response to aggression. Med Hypotheses. 2020;143:110102.
Delgado-Roche L, Mesta F. Oxidative stress as key player in severe acute respiratory syndrome coronavirus (SARS-CoV) infection. Arch Med Res. 2020;51:384–387.
Point 3: It is good to see that ACE2 has a role in Acid-mediated ALI which would be therapeutically target to address this model of human pathology. This should be highlighted in the dicussion.
Response 3: We appreciate the reviewer’s comment. ACE2 has a role in acid-mediated ALI which would be therapeutically target to address this model of human pathology.
And we revised the manuscript and highlighted the statements as followings:
“The present study showed that ACE2 reduced oxidative stress, decreased lung permeability, ameliorated pulmonary inflammation, alleviated STAT3 and ERK1/2 phosphorylation, and finally counteracted the pathogenic process of ALI. ACE2 played a role in acid-mediated ALI which would be a therapeutically target for further research.” (Lines 460-464 in the Discusssion section of revised manuscript)
“Our results indicated that ACE2 could counteract the proinflammatory effect of ALI and attenuate ALI pathogenesis. ACE2 effectively alleviates the pathogenic process of ALI by regulating MAPK, STAT3 and MMP activation” (Lines 493-497 in the Conclusion section of revised manuscript)
We have carefully revised and responded point-by-point to the comments, suggestions, and corrections. We hope you will find our revised manuscript satisfactory and suitable for publication in Medicina.
Thank you very much,
With best regard,
Chih-Sheng Lin, Ph.D.
Department of Biological Science and Technology
National Yang Ming Chiao Tung University
No.75 Po-Ai Street, Hsinchu 30068, Taiwan
Tel.: +886-3-5131338
E-mail: lincs@nycu.edu.tw

Reviewer 2 Report
The protective effects of NRICM101 in acid-induced lung injury were studied in both ACE2 tg and knockout mice. And the authors emphasize that the results showed that the pathological features of acid-induced ALI were similar to those of COVID-19-related lung damage.
Comments:
1. Sars-Cov-2 virus induced lung injury is more complex than acid-induced lung injury. The increase of inflammatory cytokines, lung permeability and leukocyte infiltration are common characters of acute lung injury. The results are not sufficient to support the conclusion that acid-induced ALI were similar to those of COVID-19-related lung damage.
2. In the Acid+TCM group, the mice were administered NRICM101 for 4 days. Form which day the mice start to be given NRICM101, Day 3?
3. Figure 3a, the authors claim that lung vascular permeability was evaluated by measuring EBD leakage. However, the EBD was instilled by intratracheal instead of through vessel? These severe leakages showed in Figure 3 can’t happened in the mice which have only 5% body weight drop (WT acid group).
4. NRICM101 showed more protective effect in ACE2 KO mice compare to WT and ACE2 tg mice, (Figure 2, 4 and 5). And the protective tendency of these three types of mice is similar, which suggested the protective effect of NRICM101 is through ACE2. The double-edged sword of ACE2 can’t explain these results, the authors should think about other mechanism.
5. Figure 6, the authors claimed acid intratracheal instillation significantly induced pulmonary ACE expression, and the level of ACE was decreased in mice administered NRICM101. However, the level of ACE was increased in mice administered NRICM101 rather than decreased in ACE2 KO mice.
6. Figure 7, total STAT3 and ERK1/2 should be shown to compare with phosphonate STAT3 and ERK1/2.
Author Response
Response to Reviewer 2 Comments
Responses to Reviewer’s comments:
We deeply appreciate the reviewer’s time and efforts to review the manuscript, “ACE2 and a Traditional Chinese Medicine Formula NRICM101 Could Alleviate the Inflammation and Pathogenic Process of Acute Lung Injury” (Medicina-2520593)”, and provide very critical and insightful comments. The comments raised were extremely helpful and have been fully integrated into the revised submission. We sincerely appreciate you for the detailed comments and suggestions to improve the value of manuscript. Each of these comments have been addressed.
The revised portions were highlighted in red in the revised manuscript. The followings are our point-by-point responses to the comments:
Point 1: Sars-Cov-2 virus induced lung injury is more complex than acid-induced lung injury. The increase of inflammatory cytokines, lung permeability and leukocyte infiltration are common characters of acute lung injury. The results are not sufficient to support the conclusion that acid-induced ALI were similar to those of COVID-19-related lung damage.
Response 1: We sincerely appreciate and agree the reviewer’s comment. Sars-Cov-2 virus induced lung injury is more complex than acid-induced lung injury. However, the challenge in studying SARS-CoV-2 pathogenicity is the limited availability of animal models because the highly infective nature of SARS-CoV-2 must be conducted in a biosafety level 3 (BSL-3) facility. In COVID-19 mediated-ALI, ACE2 depletion by SARS-CoV-2 spike protein binding disrupts the balance between ACE and ACE2 [Banu et al.,2020]. In this study, the mice with genetic modifications absence of ACE2 (mACE2 KO) is presented as ACE2 depletion by SARS-CoV-2 spike protein binding, and the acid intratracheal instillation could disrupted the balance between ACE2 and ACE in a short time and induced the inflammatory response [Imai et al., 2005]. Thus, acid-induced ALI in ACE2 modifications mice were used as an alternative to COVID-19 mediated-ALI under the safety condition. The above statements were mentioned in the Introduction section of revised manuscript. (Lines 70-74, 91-102 in the Introduction section of revised manuscript)
Banu, N.; Panikar, S.S.; Leal, L.R.; Leal, A.R. Protective role of ACE2 and its downregulation in SARS-CoV-2 infection leading to Macrophage Activation Syndrome: Therapeutic implications. Life Sci 2020, 256, 117905, doi:10.1016/j.lfs.2020.117905.
Imai Y, Kuba K, Rao S, Huan Y, Guo F, Guan B, Yang P, Sarao R, Wada T, Leong-Poi H, Crackower MA, Fukamizu A, Hui CC, Hein L, Uhlig S, Slutsky AS, Jiang C, Penninger JM. Angiotensin-converting enzyme 2 protects from severe acute lung failure. Nature. 2005 Jul 7;436(7047):112-6. doi: 10.1038/nature03712. PMID: 16001071; PMCID: PMC7094998.
Point 2: In the Acid+TCM group, the mice were administered NRICM101 for 4 days. Form which day the mice start to be given NRICM101, Day 3?
Response 2: Thank you for the valuable comment. In the Acid+TCM group, the mice start to be given NRICM101 on Day 4. According to the suggestion, we revised the manuscipt as followings:
“In the Acid+TCM group, the mice were treated with acid intratracheal instillation twice on Day 0 and Day 2 and then treated with NRICM101 by oral gavage for 4 days starting from Day 3 to Day 6 (3 g/kg of body weight/day).” (Lines 150-154 in the Materials and Methods section of revised manuscript)
“C57BL/6 mice (wild type, WT), K18-hACE2 transgenic mice (K18-hACE2 TG), and mACE2 knockout mice (mACE2 KO) were treated with PBS by intratracheal instillation twice on Day 0 and Day 2 in the Control group. The mice were treated with acid intratracheal instillation twice on Day 0 and Day 2 in the Acid group. The mice were treated with acid intratracheal instillation twice on Day 0 and Day 2 and then administered NRICM101 for 4 days from Day 3 to Day 6 in the Acid+TCM group (Table 1).” (Lines 237-242 in the Result section of revised manuscript)
Point 3: Figure 3a, the authors claim that lung vascular permeability was evaluated by measuring EBD leakage. However, the EBD was instilled by intratracheal instead of through vessel? These severe leakages showed in Figure 3 can’t happened in the mice which have only 5% body weight drop (WT acid group).
Response 3: Thank you for the pertinent comment. The EBD method of this study was performed according to the protocol previously described by Mayeux et al (2019). It is a common method that uses EBD to detect acute lung injury by intratracheal instillation [Mayeux et al., 2019]. These severe leakages shown in Figure 3, lung permeability of the WT group were significantly increased after acid treatment. And the body weight of WT mice decreased not 5%, but 10% compared with the control. According to the suggestion, we revised the manuscipt as followings:
“Lung vascular permeability was evaluated by measuring EBD leakage. It is a common method that use EBD to detect acute lung injury by intratracheal instillation [38]. The mice were treated with EBD (Santa Cruz, Dallas, TX, USA) by intratracheal instillation 1 h before being euthanized. The lungs were isolated, weighed, and incu-bated with formamide (500 μL) at 55 °C for 48 h to extract the dye. The formamide and EBD mixture was centrifuged to pellet any remaining tissue fragments. The optical density was spectrophotometrically determined at 620 nm (Thermo Scientific, Wal-tham, MA, USA).” (Lines 160-166 in the Materials and Methods section of revised manuscript)
“In the acid group, the body weight loss of WT mice, K18-hACE2 TG and mACE2 KO are 10%, 5% and 20%, respectively. Notably, mACE2 KO mice with acid intratracheal instillation exhibited a rapid and sustained declines in body weight compared with the other mice.” (Lines 245-249 in the Result section of revised manuscript)
Mayeux JM, Kono DH, Pollard KM. Development of experimental silicosis in inbred and outbred mice depends on instillation volume. Sci Rep. 2019 Oct 2;9(1):14190. doi: 10.1038/s41598-019-50725-9.
Point 4: NRICM101 showed more protective effect in ACE2 KO mice compare to WT and ACE2 tg mice, (Figure 2, 4 and 5). And the protective tendency of these three types of mice is similar, which suggested the protective effect of NRICM101 is through ACE2. The double-edged sword of ACE2 can’t explain these results, the authors should think about other mechanism.
Response 4: Thank you for the valuable comment. According to our results, NRICM101 showed a more protective effect in ACE2 KO mice compared to WT and K18-hACE2 TG mice through contouring ACE expression and ameliorating inflammation. Meanwhile, ACE2 KO mice showed serious immunocyte infiltration and alveolar injury after acid treatment. Activation of the ACE2-Ang-(1-7)/Mas receptor axis is absent in mACE2 KO mice, which exhibited severe injuries and imbalance in the RAS system. The robust increases in ACE of mACE2 KO mice induced by acid-induced ALI resulted in an imbalance between ACE and ACE2. NRICM101 could protect acid-induced ALI by decreasing the inflammatory effect in mACE2 KO mice. However, this present study didn’t show solid evidence that NRICM101 could alleviate ALI via reversing the ACE2 expression. It would require further exploration. Our results only indicated that ACE2 provided a more protective effect on lung damage; nevertheless, ACE2 is also a SARS-CoV-2 entry receptor, which suggests that the increase in ACE2 expression may lead to an increase in cell infection [Samavati & Uhal, 2020]. This has been termed the double-edged sword of ACE2. According to the suggestion, we revised the manuscipt as followings:
“Activation of the ACE2-Ang-(1-7)/Mas receptor axis is absent in mACE2 KO mice, which exhibited severe injuries and imbalance in the RAS system compared with K18-hACE2 TG mice [Imai et al, 2005]. Additionally, an increase in ACE is associated with ALI. Based on these findings, robust increases in ACE protein expression were measured in acid-induced ALI. The upregulation of ACE and depletion of ACE2 could result in an imbalance in ACE/ACE2 [Beyerstedt et al, 2021; Yang et al, 2017; Lin et al,2018], such as increasing the ACE/ACE2 ratio to accelerate the disease progression of acid-induced ALI or COVID-19 ALI.” (Lines 434-441 in the Discussion section of revised manuscript)
Imai, Y.; Kuba, K.; Rao, S.; Huan, Y.; Guo, F.; Guan, B.; Yang, P.; Sarao, R.; Wada, T.; Leong-Poi, H.; et al. Angiotensin-converting enzyme 2 protects from severe acute lung failure. Nature 2005, 436, 112-116, doi:10.1038/nature03712.
Beyerstedt, S.; Casaro, E.B.; Rangel, E.B. COVID-19: angiotensin-converting enzyme 2 (ACE2) expression and tissue susceptibility to SARS-CoV-2 infection. Eur J Clin Microbiol Infect Dis 2021, 40, 905-919, doi:10.1007/s10096-020-04138-6.
Yang, C.W.; Lu, L.C.; Chang, C.C.; Cho, C.C.; Hsieh, W.Y.; Tsai, C.H.; Lin, Y.C.; Lin, C.S. Imbalanced plasma ACE and ACE2 level in the uremic patients with cardiovascular diseases and its change during a single hemodialysis session. Ren Fail 2017, 39, 719-728, doi:10.1080/0886022X.2017.1398665.
Lin, C.I.; Tsai, C.H.; Sun, Y.L.; Hsieh, W.Y.; Lin, Y.C.; Chen, C.Y.; Lin, C.S. Instillation of particulate matter 2.5 induced acute lung injury and attenuated the injury recovery in ACE2 knockout mice. Int J Biol Sci 2018, 14, 253-265, doi:10.7150/ijbs.23489.
Samavati L, Uhal BD. ACE2, Much More Than Just a Receptor for SARS-COV-2. Front Cell Infect Microbiol. 2020 Jun 5;10:317. doi: 10.3389/fcimb.2020.00317.
Point 5: Figure 6, the authors claimed acid intratracheal instillation significantly induced pulmonary ACE expression, and the level of ACE was decreased in mice administered NRICM101. However, the level of ACE was increased in mice administered NRICM101 rather than decreased in ACE2 KO mice.
Response 5: Thank you for the value comments. The level of ACE was decreased in mACE2 KO mice administered NRICM101. Because there was an error in the privous description of the content, we revised the manuscript as followings:
“To identify the RAS mediators associated with acid-induced ALI, the protein ex-pression of ACE was investigated. In the Acid group, ACE expression in WT, K18-hACE2 TG, and mACE2 KO mice was increased by approximately 2.0-, 1.2- and 6-fold, respectively, compared with that in the Control group. There was no change in ACE expression in K18-hACE2 TG mice with acid intratracheal instillation. In the Acid+TCM group, ACE expression in WT and mACE2 KO mice was decreased 75% and 70%, respectively, compared with that in the Acid group (Figure 6).” (Lines 344-350 in the Result section of revised manuscript)
Point 6: Figure 7, total STAT3 and ERK1/2 should be shown to compare with phosphonate STAT3 and ERK1/2.
Response 6: Thank you for the pertinent comments. We have evaluated the total STAT3 and ERK1/2 and normalized with b-actin. Because there were no significantly dfifference in the experiment, the results were not included in the original manuscript. According to the comment, the total STAT3 and ERK1/2 provided in the revised version. (Figure 7 in the Result section of revised manuscript)
We have carefully revised and responded point-by-point to the comments, suggestions, and corrections. We hope you will find our revised manuscript satisfactory and suitable for publication in Medicina.
Thank you very much,
With best regard,
Chih-Sheng Lin, Ph.D.
Department of Biological Science and Technology
National Yang Ming Chiao Tung University
No.75 Po-Ai Street, Hsinchu 30068, Taiwan
Tel.: +886-3-5131338
E-mail: lincs@nycu.edu.tw

Round 2
Reviewer 2 Report
After modification, the fatal mistakes are still there.
1. The authors use reference [38] to support their method of lung vascular permeability, in this literature, Evans Blue instillation through transoral/oropharyngeal is to visualize the effect of volume on dispersal of aspirated fluid. There is nothing related to vascular permeability.
2. NRICM101 was given to mouse by oral gavage for 4 days starting from Day 3 to Day 6. The body weight of acid group and acid+TCM group are significantly different at Day2 even before NRICM101 given. (Fgure2a)
3. TGFb1 is mostly identified as anti-inflammatory cytokine, which resolves lung inflammation and causes lung fibrosis. The authors identified TGFb1 as inflammatory cytokine in acid induce ARDS model, only one reference [56] about lung fibrosis is not convinced.
Author Response
Response to Reviewer 2 Comments
Responses to Reviewer’s comments:
Thank you very much for your reviewing process of our manuscript, “ACE2 and a Traditional Chinese Medicine Formula NRICM101 Could Alleviate the Inflammation and Pathogenic Process of Acute Lung Injury” (Medicina-2520593)”.The criticisms raised are extremely helpful and have been fully responded and integrated into this revised submission. We deeply appreciate the reviewer’s detailed comments to improve the readability of the manuscript. Here we present a point-by-point response to the reviewers’ comments and concerns.
The revised portions were highlighted in red in the revised manuscript. The followings are our point-by-point responses to the comments:
Point 1: The authors use reference [38] to support their method of lung vascular permeability, in this literature, Evans Blue instillation through transoral/oropharyngeal is to visualize the effect of volume on dispersal of aspirated fluid. There is nothing related to vascular permeability.
Response 1: We sincerely appreciate and agree with the reviewer’s comment. The reference [Mayeux JM et al., 2019] which was described in our article, only showed that Evan blue dye (EBD) is a common method to detect acute lung injury by intratracheal instillation, but nothing related to vascular permeability. As reviewer’s comment, the EBD should be injected into the vein instead of intratracheal instillation to assess the pulmonary vascular leakage. Our instilled EBD results indicated lung damage instead of vascular permeability. To be more specific and concise the statement, we removed all the term “vascular permeability”, and revised the manuscript as followings:
“Evan blue dye is an indicator of acute lung damage. It is a common method that use EBD to detect acute lung injury by intratracheal instillation [Kim et al., 2009; Sudhadevi et al., 2021; Pelgrim et al., 2022].” (Lines 160-161 in the revised manuscript)
“3.3 ACE2 and NRICM101 decreased lung damage in ALI mice induced by acid instillation” (Line 275 in the revised manuscript)
“Evans blue dye (EBD) was used to investigate the lung damage.” (Line 276 in the revised manuscript)
Kim, J.S.; Lee, B.; Hwang, I.C.; Yang, Y.S.; Yang, M.J.; Song, C.W. An automatic video instillator for intratracheal instillation in the rat. Lab Anim 2010, 44, 20-24, doi:10.1258/la.2009.009003.
Sudhadevi, T.; Ha, A.W.; Harijith, A. A Minimally Invasive Method for Intratracheal Instillation of Drugs in Neonatal Rodents to Treat Lung Disease. J Vis Exp 2021, doi:10.3791/61729.
Pelgrim, C.E.; van Ark, I.; Leusink-Muis, T.; Brans, M.A.D.; Braber, S.; Garssen, J.; van Helvoort, A.; Kraneveld, A.D.; Folkerts, G. Intratracheal administration of solutions in mice; development and validation of an optimized method with improved efficacy, reproducibility and accuracy. J Pharmacol Toxicol Methods 2022, 114, 107156, doi:10.1016/j.vascn.2022.107156.
Point 2: NRICM101 was given to mouse by oral gavage for 4 days starting from Day 3 to Day 6. The body weight of acid group and acid+TCM group are significantly different at Day2 even before NRICM101 given. (Fgure2a)
Response 2: Thank you for the pertinent comment. There are no significantly difference on the body weight at Day2 between the acid group and the acid+TCM group (p= 0.2346). The body weight of acid group and acid+TCM group are 80% and 84%, respectively, compared with those in the Control group.
Point 3: TGFb1 is mostly identified as anti-inflammatory cytokine, which resolves lung inflammation and causes lung fibrosis. The authors identified TGFb1 as inflammatory cytokine in acid induce ARDS model, only one reference [56] about lung fibrosis is not convinced.
Response 3: We sincerely appreciate and agree the reviewer’s comment. TGF-β1 is identified as anti-inflammatory cytokine. However, TGF-β1 is also involved in the mechanism of many diseases, playing a dual role of inhibiting or promoting inflammatory diseases. In this study, acid intratracheal instillation in mice induced an inflammatory response and induced the production of TGF-β1 in the lungs. Some studies have shown that TGF-β1 plays a major role under inflammatory conditions [Jiang et al., 2003]. TGF-β1 in the presence of IL-6 drives the differentiation of T helper 17 (Th17) cells, which can promote further inflammation and augment autoimmune conditions [Sanjabi et al., 2009]. This capacity of TGF-β1 to induce either immunosuppressive or inflammatory events is context dependent, and must be considered when analyzing its role in disease [Prud'homme, 2007]. Because TGF-β1 is a controversial role of inhibiting or promoting inflammation, we decided to remove all statements about TGF-β1 from the manuscript, and revised the results Figure 5 as followings:
(Figure 5 in the revised manuscript, the TGF-β1 result had been removed. )
Sanjabi, S.; Zenewicz, L.A.; Kamanaka, M.; Flavell, R.A. Anti-inflammatory and pro-inflammatory roles of TGF-beta, IL-10, and IL-22 in immunity and autoimmunity. Curr Opin Pharmacol 2009, 9, 447-453, doi:10.1016/j.coph.2009.04.008.
Jiang, Z.; Seo, J.Y.; Ha, H.; Lee, E.A.; Kim, Y.S.; Han, D.C.; Uh, S.T.; Park, C.S.; Lee, H.B. Reactive oxygen species mediate TGF-beta1-induced plasminogen activator inhibitor-1 upregulation in mesangial cells. Biochem Biophys Res Commun 2003, 309, 961-966, doi:10.1016/j.bbrc.2003.08.102.
Prud'homme, G.J. Pathobiology of transforming growth factor beta in cancer, fibrosis and immunologic disease, and therapeutic considerations. Lab Invest 2007, 87, 1077-1091, doi:10.1038/labinvest.3700669.
We truly hope that the revised manuscript has significantly been improved toward Medicina standards. We look forward to hearing from you about our submission in due course and answering any further questions you may have.
Thank you very much,
Best regards,
Chih-Sheng Lin, Ph.D.
Department of Biological Science and Technology
National Yang Ming Chiao Tung University
No.75 Po-Ai Street, Hsinchu 30068, Taiwan
Tel.: +886-3-5131338
E-mail: lincs@nycu.edu.tw

Round 3
Reviewer 2 Report
I checked the three references the author mentioned, Evans blue dye isn’t used to show the lung injury but to demonstrate the accuracy of the delivery method in all three literatures.
If the same amounts of Evans blue dye were given to control group (non-injury lung) and acid group (injury lung) by intratracheal instillation, how the injuries were showed by the dye? In Figure 3a, the control group showed significantly less stain comparing to acid group, where the dye goes from lung alveoli in 1hour?
Author Response
Response to Reviewer 2 Comments
Responses to Reviewer’s comments:
We deeply appreciate the reviewer’s time and efforts to review the manuscript, “ACE2 and a Traditional Chinese Medicine Formula NRICM101 Could Alleviate the Inflammation and Pathogenic Process of Acute Lung Injury” (Medicina-2520593)”, and provide very critical and insightful comments. The comments raised were extremely helpful and have been fully integrated into the revised submission. We sincerely appreciate you for the detailed comments and suggestions to improve the value of manuscript. Each of these comments have been addressed.
The revised portions were highlighted in red in the revised manuscript. The followings are our point-by-point responses to the comments:
Point 1: I checked the three references the author mentioned, Evans blue dye isn’t used to show the lung injury but to demonstrate the accuracy of the delivery method in all three literatures.
If the same amounts of Evans blue dye were given to control group (non-injury lung) and acid group (injury lung) by intratracheal instillation, how the injuries were showed by the dye? In Figure 3a, the control group showed significantly less stain comparing to acid group, where the dye goes from lung alveoli in 1hour?
Response 1: We sincerely appreciate and agree with the reviewer’s comment. The reference [Kim et al., 2009; Sudhadevi et al., 2021; Pelgrim et al., 2022] which were described in our article, only showed that Evan blue dye (EBD) is the accuracy of the delivery method by intratracheal instillation, but not to show the lung injury.
According to previous studies, albumin has been found to penetrate barriers and accumulate in airspaces in response to acute lung injury and thus was used as a marker for lung barrier failure (Moitra et al., 2007; Yao et al., 2018). Because EBD could bind to serum albumin and have high water solubility, EBD is used in biomedicine including the estimation of blood volume, the assessment of vascular permeability.
Acid-induced acute lung injury resulted in a greater degree of extravasation of the EBD in the lung tissue. In the study, EBD-albumin conjugate had circulated for 1 h. If EBD would not bind to albumin, EBD was washed after animal perfusion. As reviewer’s comment, the EBD should use to detect acute lung injury by intravenous injection instead of intratracheal instillation to assess the pulmonary vascular leakage. To make sure the statements are as specific and accurate as possible, we decided to remove all statements and data about EBD in last the manuscript.
- Moitra, J.; Sammani, S.; Garcia, J.G. Re-evaluation of Evans Blue dye as a marker of albumin clearance in murine models of acute lung injury. Translational Research 2007, 150, 253-265.
- Yao, L.; Xue, X.; Yu, P.; Ni, Y.; Chen, F. Evans Blue Dye: A Revisit of Its Applications in Biomedicine. Contrast Media Mol Imaging 2018, 2018, 7628037, doi:10.1155/2018/7628037.
We truly hope that the revised manuscript has significantly been improved toward Medicina standards.
Best regards
Chih-Sheng Lin, Ph.D.
Department of Biological Science and Technology
National Yang Ming Chiao Tung University
No.75 Po-Ai Street, Hsinchu 30068, Taiwan
Tel.: +886-3-5131338
E-mail: lincs@nycu.edu.tw
